H2-saturation of high affinity H2-oxidizing bacteria alters the ecological niche of soil microorganisms unevenly among taxonomic groups

Piché-Choquette Sarah 1
Tremblay Julien 2
Tringe Susannah G. 3
Constant Philippe 1 philippe.constant@iaf.inrs.ca
1 INRS-Institut Armand-Frappier , Laval, Quebec , Canada
2 Biomonitoring, National Research Council Canada , Montreal, Quebec , Canada
3 DOE Joint Genome Institute , Walnut Creek, California , United States of America
Corbo Maria Rosaria
Electronic publication date: 2016 Mar 10
Publication date: 2016
Volume: 4
Electronic Location ID: e1782
Received 2016 Jan 13; Accepted 2016 Feb 17
Copyright: ©2016 Piché-Choquette et al.
Copyright year: 2016
Copyright holder: Piché-Choquette et al.
License: This is an open access article distributed under the terms of the Creative Commons Attribution License, which permits unrestricted use, distribution, reproduction and adaptation in any medium and for any purpose provided that it is properly attributed. For attribution, the original author(s), title, publication source (PeerJ) and either DOI or URL of the article must be cited.
License URL: https://creativecommons.org/licenses/by/4.0/

Keywords: Soil, Microbial ecology, Correlation network

Funding: Natural Sciences and Engineering Research Council of Canada Discovery Community Sequencing Program of the Joint Genome Institute (US Department of Energy) US Department of Energy Joint Genome Institute DOE Office of Science User Facility Office of Science of the US Department of Energy DE-AC02-05CH11231 This work has been supported by a Natural Sciences and Engineering Research Council of Canada Discovery grant to PC and by the Community Sequencing Program of the Joint Genome Institute (US Department of Energy) to PC, JT and SGT. The work conducted by the US Department of Energy Joint Genome Institute, a DOE Office of Science User Facility, is supported by the Office of Science of the US Department of Energy under Contract No. DE-AC02-05CH11231. The funders had no role in study design, data collection and analysis, decision to publish, or preparation of the manuscript.

==============================
Soil microbial communities are continuously exposed to H2 diffusing into the soil from the atmosphere. N2-fixing nodules represent a peculiar microniche in soil where H2 can reach concentrations up to 20,000 fold higher than in the global atmosphere (0.530 ppmv). In this study, we investigated the impact of H2 exposure on soil bacterial community structure using dynamic microcosm chambers simulating soil H2 exposure from the atmosphere and N2-fixing nodules. Biphasic kinetic parameters governing H2 oxidation activity in soil changed drastically upon elevated H2 exposure, corresponding to a slight but significant decay of high affinity H2-oxidizing bacteria population, accompanied by an enrichment or activation of microorganisms displaying low-affinity for H2. In contrast to previous studies that unveiled limited response by a few species, the relative abundance of 958 bacterial ribotypes distributed among various taxonomic groups, rather than a few distinct taxa, was influenced by H2 exposure. Furthermore, correlation networks showed important alterations of ribotype covariation in response to H2 exposure, suggesting that H2 affects microbe-microbe interactions in soil. Taken together, our results demonstrate that H2-rich environments exert a direct influence on soil H2-oxidizing bacteria in addition to indirect effects on other members of the bacterial communities.

Introduction

Soil microbial communities are continuously exposed to molecular hydrogen (H2). Trace levels of H2 (0.530 ppmv) diffuse into the soil from the global atmosphere, yet higher concentrations can be found in the rhizosphere of N2-fixing legumes (Constant, Poissant & Villemur, 2009). Indeed, H2 is an obligate by-product of nitrogenase in N2-fixing free living or symbiotic bacteria, with an H2 molecule produced for every reduced N2 molecule (Hoffman, Dean & Seefeldt, 2009). It has been estimated that 240,000 L of H2 are produced per hectare of legume crop over a growing season (Dong et al., 2003). At the microscale level, there is a steep H2 concentration gradient starting at about 20,000 ppmv at the soil-nodule interface and ending with sub-atmospheric levels a few centimeters away (La Favre & Focht, 1983; Rasche & Arp, 1989; Witty, 1991). Despite the high concentrations of H2 found in legumes rhizosphere, a very small proportion escapes to the atmosphere due to H2-oxidizing bacteria (HOB) thriving in soil. These microorganisms also play a vital role in the global budget of H2, being responsible for about 80% (60 Tg H2 yr−1) of the global losses of this trace gas from the atmosphere (Constant, Poissant & Villemur, 2009; Ehhalt & Rohrer, 2009).

Aerobic soil bacteria scavenging H2 diffusing from the atmosphere and N2-fixing nodules encompass a broad range of taxonomic groups. These microorganisms generally possess one or up to four different types of [NiFe]-hydrogenases catalyzing the interconversion of H2 into protons and electrons (H2↔2H+ + 2e−). These hydrogenases are classified into four distinct phylogenetic groups, each displaying particular physiological roles (Constant & Hallenbeck, 2012; Vignais & Billoud, 2007). Hydrogenases encompassing group 1h are generally characterized by a high affinity towards H2( (app)Km < 100ppmv), conferring the ability to oxidize atmospheric H2 (Constant et al., 2010). Genome database mining of hhyL gene encoding the large subunit of high affinity hydrogenases are differentially distributed in Actinobacteria, with few representatives of Proteobacteria, Acidobacteria and Chloroflexi (Constant et al., 2011a; Constant et al., 2011b). The energy yield associated with the oxidation of atmospheric H2 is insufficient to support chemolithotrophic growth and compelling experimental evidence suggest that H2 supplies maintenance energy requirements and mixotrophic growth in high affinity HOB (Constant et al., 2011a; Constant et al., 2011b; Constant et al., 2010; Greening et al., 2014a; Greening et al., 2015). The distribution of the other [NiFe]-hydrogenase groups is broader, including methanogenic archaea and Cyanobacteria (Vignais & Billoud, 2007). Ralstonia eutropha and Bradyrhizobium japonicum, part of the knallgas bacteria functional group, are able to scavenge H2 diffusing from nodules in the presence of O2, yet they are unable to use atmospheric H2 due to the low-affinity ( (app)Km > 1,000ppmv) and high H2 threshold concentration of their [NiFe]-hydrogenases (Conrad, Aragno & Seiler, 1983). Knallgas bacteria can use H2 as a sole or supplementary energy source in chemolithotrophic or mixotrophic growth, respectively. The co-occurrence of these two sub-populations of HOB, as defined by substrate affinity, is supported by the biphasic kinetics governing H2 oxidation activity in soil (Häring & Conrad, 1994).

Laboratory incubations simulating H2 fluxes from N2-fixing nodules demonstrated that soil bacterial community composition changed upon H2 exposure (Osborne, Peoples & Janssen, 2010; Stein et al., 2005; Zhang, He & Dong, 2009). However, methods previously used provided a low taxonomic resolution and coverage of bacterial communities responding to H2 exposure. Here, we revisited these experiments using a combination of high-throughput sequencing of the bacterial 16S rRNA gene and an H2 metabolism analysis to compare microbial community structure in dynamic microcosm chambers simulating soil H2 exposure from the atmosphere and from N2-fixing nodules, corresponding to unsaturating and saturating H2 concentrations for high affinity hydrogenases, respectively. Elevated H2 exposure was expected to activate and enrich HOB, consequently leading to indirect impacts on the whole soil bacterial community through competitive and synergistic microbe-microbe interactions.

Materials and Methods

Soil sample

Soil sample was collected in an agricultural land located in St. Claude (Québec, Canada) on the south shore of the St. Lawrence River (45.6809°N, −71.9969°W). The field is managed with fallow, potatoes and maize crop rotation. Potato seedlings (approximately 10 cm height) were present on the site during sampling in July 2013. The top layer of the A-horizon (0–10 cm depth) was collected, stored in plastic bags at 4 °C and processed within a week. Soil was air-dried for 48 h in the laboratory and sieved (2 mm mesh size) through a vibratory sieve shaker AS 200® (Retsch GmbH, Haan, Germany) before preparation of soil microcosms. Soil was classified as sandy clay loam according to soil textural class parameters identified with the hydrometer method (Bouyoucos, 1936). Soil pH was determined with 1:2 soil-water suspensions with an Accumet pH-meter (Fisher Scientific, Hampton, NH). Total carbon (3.1 ± 0.3%) and total nitrogen (0.3 ± 0.0%) content were determined using an elemental combustion system using the protocol described in Khdhiri et al. (2015).

Controlled H2 exposure in dynamic soil microcosm chambers

Dynamic microcosm chambers were designed to expose soil to controlled levels of H2 (Fig. S1). Microcosm chambers consisted of 0.9 L cell culture flasks (Corning, Tewksbury, MA, USA) equipped with a rubber stopper fitted with two 1/8” outside diameter PTFE (Teflon) tubes: the first supplied gas mixture to the microcosm chamber and the second was vented to the atmosphere. Synthetic gas mixtures supplied the microcosm chambers at a flowrate of 40 ml min−1, resulting in a dynamic headspace with a residence time of approximately 22 min in the enclosures. Gas mixtures were bubbled in water before entering in microcosms to prevent soil dryness. Two different H2 treatments were applied in parallel incubations. The first treatment (designated microcosms eH2(a) and eH2(b)), named elevated H2 (eH2), consisted of exposing soil to a dynamic headspace comprising 525 ppmv H2 in synthetic air, simulating H2 concentrations detected around N2-fixing nodules in soil (Hunt & Layzell, 1993). The second was a control treatment (designated microcosms aH2(a) and aH2(b)), named atmospheric H2 (aH2), where soil was exposed to a dynamic headspace comprising 0.54 ppmv H2 in synthetic air, representing H2 concentrations found in the atmosphere. Both treatments were replicated, resulting in four soil microcosms in total. Each microcosm chamber contained 200 g(dw) soil at the beginning of the incubation period. This amount of soil ensured sufficient material for the monitoring of microbiological and physicochemical variables throughout the incubation period, while avoiding diffusion limitation of the H2 soil uptake rate measurements. Diffusion limitation was precluded since preliminary experiments using the same microcosm setup showed proportional H2 uptake activity as a function of the amount of soil in the chamber using 150, 200 and 250 g(dw) soil samples (raw data file). Soil water content was adjusted to 20% water holding capacity before incubation for 10 days at 28 °C in the dark. Synthetic gas mixture supply was continuously maintained, with the exception of routine soil subsamples collection and high affinity H2 oxidation rate measurements (see below). A decrease in soil pH was observed over the course of the incubation (from pH 5.9 ± 0.2 to 5.1 ± 0.1) in all microcosms, without distinction between aH2 and eH2 treatments. Soil moisture was monitored using standard gravimetric method and maintained at 26 ± 5% throughout the incubation period. Blank (empty) microcosms incubated prior to the experiment did not show any H2 oxidation or production activity.

H2 uptake activity

High affinity H2 oxidation activity was routinely monitored throughout the incubation of the soil microcosms. Briefly, microcosms were disconnected from their respective gas supply, flushed 5 min with a synthetic gas mixture (0.54 ppmv H2) and tightly closed with rubber septum caps. The 5-min flush was shown sufficient to avoid residual H2 degassing that would otherwise have led to an underestimation of H2 oxidation rate. A defined volume of H2 gas mixture (525 ± 10 ppmv H2 GST-Welco, PA, USA.) was injected to obtain an initial concentration of approximately 3 ppmv in the static headspace. Decrease of the H2 mixing ratio was monitored as a function of time by analyzing aliquots (10 ml) of the headspace air in a ta3000R gas chromatograph equipped with a reduction gas detector (Ametek Process Instruments®, DE, USA.) as previously described (Khdhiri et al., 2015). Considering the low level of H2 added in the headspace, H2 uptake reflected the activity of high affinity HOB, knallgas bacteria showing low affinity for H2 being unable to use these trace amounts of H2 (Conrad, Aragno & Seiler, 1983). The biphasic kinetic parameters governing H2 oxidation activity in soil (i.e., (app)Km and (app)Vmax) were measured at the end of the 10-day incubation period after the addition of specified amounts of pure H2 into the headspace of the microcosms, as previously described (Schuler & Conrad, 1990).

DNA extraction, qPCR and high-throughput sequencing of PCR-amplified bacterial 16S rRNA gene

Soil subsamples (approximately 10 g per subsample, without replacement) were collected in the four microcosms after 0, 1, 3, 5, 7 and 10 incubation days to investigate the taxonomic structure of microbial communities. The extraction of total genomic DNA was performed using the FastDNA SPIN kit for Soil® (MP Biomedicals, Solon, OH, USA.). DNA was eluted in 100 µL nuclease-free water. Quality of the DNA was then examined on agarose gels and samples were quantified using Quantifluor dsDNA System® (Promega, Fitchburg, WI, USA). The V4 region of bacterial 16S rRNA gene was PCR amplified (Table S1) and sequenced on an Illumina MiSeq 2000 instrument as multiplexed libraries to generate paired-end reads (2 × 250 bp). Quantification of bacterial 16S rRNA and hhyL genes was done by qPCR following Khdhiri et al. (2015).

Quality control of sequencing reads

Our sequencing analysis pipeline using QIIME 1.8.0 (Kuczynski et al., 2011) software was based on the Itagger pipeline (Tremblay et al., 2015). V4 region Illumina sequencing reads were first processed using the split_library function implemented in QIIME for samples demultiplexing and forward primer removal. The assembly of paired-end reads was performed using FLASH software version 1.2.10 (Magoc & Salzberg, 2011). The software DUK (Li, Copeland & Han, 2011), a Kmer-based sequence matching tool, was then used to remove common sequencing contaminants as well as PhiX sequences added as control. Afterwards, sequences were trimmed to remove low quality ends and a general quality control was also applied. All reads containing at least a single N (ambiguous base), an average quality score of less than 30 or more than 15 bases with a quality score under 20 were discarded. The minimum and maximum read length were respectively set to 200 and 300 bases for the assembled paired-reads. Reverse primers were also removed using the truncate_reverse_primer function implemented in QIIME. Raw sequences were deposited in the Sequence Read Archive of the National Center for Biotechnology Information under the Bioproject PRJNA295403.

Clustering and taxonomic identification of sequencing reads

Processed 2,554,543 high quality reads were clustered into OTUs (Operational Taxonomic Units) using the function pick_otus implemented in QIIME along with the USEARCH (Edgar, 2010) clustering algorithm. Sequences dereplication was performed at 100% sequence identity and clusters were formed by denoising unique sequences at 99% identity. Singletons were removed to avoid diversity bias. Remaining clusters were filtered using UCHIME (Edgar et al., 2011) chimera filter in de novo mode, followed by a reference-based filter step against the Gold reference database. These 2 chimera scanning steps allow a better removal of chimeras by removing sequences flagged as chimeras by any of the two methods. Chimera-checked clusters were then clustered into OTUs at 97% identity threshold also using the function pick_otus with the USEARCH software. Taxonomic identification of OTUs was performed using the assign_taxonomy function along with the naïve Bayesian Ribosomal Database Project (RDP) classifier 2.7 (Wang et al., 2007). Taxonomy was assigned based on the Greengenes taxonomy with the 16S rRNA Greengenes reference database (DeSantis et al., 2006). A rarefaction step was applied to the OTU libraries to standardize all libraries, by random subsampling, to the lowest amount of sequences (74,316 reads) to avoid bias introduced due to unequal sequencing efforts of the samples. This rarefied OTU table comprising the frequency distribution of the OTUs in each sample was used in downstream statistical and co-occurrence network analyses.

Statistical analyses

Statistical analyses were performed using the software R version 3.0.2 (R Development Core Team, 2008). Pairwise comparison of H2 uptake activities in soil microcosms incubated under aH2 and eH2 exposure treatments was tested using one-way analysis of variance (ANOVA) and post hoc Tukey test. Shapiro–Wilk normality test was applied to confirm normal distribution of data before the ANOVA. The package “nlme” was used to compute nonlinear regression for modeling the first-order decay of hhyL gene abundance determined by qPCR. Discrimination of the samples according to their ribotyping profile was performed with the package “vegan.” Hellinger transformation of the OTU table (97% identity threshold) was computed due to the presence of zeros in the OTU table (Legendre & Gallagher, 2001). Euclidean distance matrix was used to generate an UPGMA agglomerative clustering of the samples. Identification of statistically different clusters was done by performing 999 permutations of the OTU table dataset separately across the samples and comparing the observed similarity score of each cluster against the expected values under the null hypothesis using the similarity profile tool (SIMPROF) implemented in the package “clustsig” 1.1 (Clarke, Somerfield & Gorley, 2008). Pairwise comparison of relative abundance of OTUs having a higher contribution than average to explain the two dimensions of the PCA space in soil microcosms incubated under aH2 and eH2 exposure treatments was tested using Kruskal–Wallis and post hoc Tukey test. Pairwise comparison of relative abundance of all OTUs in soil microcosms incubated under aH2 and eH2 exposure treatments was tested using the likelihood ratio test using the package “edgeR.” Covariation among OTUs during the incubation under controlled H2 levels was analyzed by correlation networks using the package “WGCNA” 1.41 (Langfelder & Horvath, 2008). A detailed methodology for the computation of correlation networks is provided in Text S1.

Figure 1 High affinity H2 oxidation rates.

Time series of the high-affinity H2 oxidation rate measured in soil microcosms exposed to aH2 or eH2 throughout the incubation period.

Figure 2 Time series of qPCR data.

Time series of (A) hhyL and (B) 16S rRNA gene abundance in soil as determined by qPCR.

Results and Discussion

Impact of H2 exposure on the distribution and activity of HOB

Agricultural soil microcosms were incubated under two different H2 level treatments. The first treatment simulated soil exposure to atmospheric level of H2 (aH2; 0.54 ppmv), while the second simulated high affinity hydrogenases substrate saturation (eH2; 525 ppmv). High affinity H2 oxidation rates were not significantly different (ANOVA, P > 0.05) between the two pairs of microcosms at the beginning of the incubation (Fig. 1). The influence of H2 exposure could already be observed after 24 h. From hour 24 to day 10, a decline of high affinity H2 uptake rate was observed in microcosms exposed to eH2, while an increase of this uptake rate was measured in microcosms exposed to aH2 levels. This trend was maintained over the course of the incubation, with an oxidation rate of 6.2 ± 1.9 and 0.3±0.1nmolgdw−1h−1 in aH2 and eH2 treatments at the end of the experiment, respectively. This response of H2 oxidation rate was accompanied by an alteration in the abundance of presumptive high affinity HOB in soil. The abundance of hhyL gene determined by qPCR did not change over the course of the incubation under aH2 (exponential regression, P = 0.06), while a slight but significant exponential decay (exponential regression, P = 0.02) was observed under eH2 (Fig. 2A). Indeed, the physiological role of high affinity hydrogenases differs within various taxonomic groups of bacteria. In streptomycetes, the enzyme is primarily expressed in spores to support a seed bank under a mixotrophic survival energy mode (Constant et al., 2010; Constant, Poissant & Villemur, 2008; Liot & Constant, 2015), while Mycobacterium express the enzyme in the exponential and stationary phases for mixotrophic growth as well as survival (Berney & Cook, 2010; Greening et al., 2015b; Greening et al., 2014b). Under axenic cultivation conditions, eH2 level would be expected to favour an increase of Mycobacterium biomass and persistence of streptomycetes spores. It should be noted that our qPCR data cannot differentiate between active and inactive cells but the decreasing trend of hhyL copy number in soil exposed to eH2 suggests that H2 was not sufficient to promote growth and persistence of high affinity HOB.

Figure 3 Kinetic parameters governing H2 oxidation activity in soil microcosms incubated under (A) aH2 and (B) eH2 exposure.

Michaelis-Menten graphs are presented with Eadie-Hofstee plots highlighting the biphasic kinetic of the reaction in the inserts.

H2 soil exposure exerted a significant impact on the kinetic parameters governing H2 oxidation activity in soil. At the end of the incubation, microcosms exposed to aH2 level displayed (app)Km of 40 ± 5 ppmv (Fig. 3A), while microcosms exposed to eH2 showed lower affinity towards H2 with an (app)Km of 838 ± 163 ppmv (Fig. 3B). Similar observations were obtained by Dong & Layzell (2001), where soil exposed to elevated levels of H2 (600 ppmv) displayed an (app)Km of 1,028 ppmv H2, while soil exposed to low H2 level (0.55 ppmv) was characterized by an (app)Km of 40 ppmv. The biphasic kinetic parameters characteristics of low and high affinity HOB were also computed using Eadie-Hofstee plots to highlight high- and low-affinity H2 uptake activities in both H2 treatments. The H2 oxidation activity was mainly catalyzed by bacteria demonstrating intermediate and high-affinity in aH2 treatment (Fig. 3A, insert), while H2-oxidation activity by bacteria displaying low- and high-affinity for H2 were detectable in the eH2 treatment (Fig. 3B, insert). Low H2 exposure supported the metabolism of intermediate- to high-affinity HOB, such as some Actinobacteria species of Streptomyces, Rhodococcus and Mycobacterium (Berney et al., 2014; Constant et al., 2011b; Golding et al., 2012; Liot & Constant, 2015; Meredith et al., 2014; Schäfer, Friedrich & Lenza, 2013). Triggering of low-affinity H2 oxidation activity under eH2 treatment might be explained by knallgas bacteria such as Proteobacteria species encompassing Ralstonia, Variovorax and Bradyrhizobium using H2 for mixotrophic growth (Rittenberg & Goodman, 1969), as they are known to exhibit a H2 uptake threshold ranging from 1 to 200 ppmv H2 (Conrad, 1996).

Impact of H2 exposure on soil bacterial community structure

Soil subsamples were collected in the microcosms during the incubation period to evaluate the temporal variation of soil bacterial community structure. H2 treatment did not influence the abundance of 16S rRNA gene in soil (Fig. 2B). This is in contrast with another study that reported more than twofold increase of the amount of bacterial cells upon elevated H2 exposure, but a mistake in the reported H2 concentration units by the authors (i.e., the use of a gas mixture of 2,000 ml H2 per L of synthetic air) impairs a sound comparison with the present investigation (Stein et al., 2005). High-throughput sequencing of PCR-amplified 16S rRNA gene unveiled that bacterial communities were dominated by Proteobacteria (34%), Acidobacteria (20%), Actinobacteria (10%), Verrucomicrobia (8%) and Bacteroidetes (5%) (Fig. S2). Species richness was not affected by H2 exposure (ANOVA, P > 0.05), with Shannon indices of 9.38 ± 0.05 and 9.34 ± 0.17 for aH2 and eH2 treatment respectively. An UPGMA hierarchical clustering analysis was computed to compare bacterial community profiles from soil subsamples collected in microcosms exposed to different H2 concentrations. The four microcosms encompassed the same cluster before the incubation, indicating high similarity of their initial ribotyping profile (Fig. 4A). No coherent impact of H2 exposure on microbial community structure was observed after 1 and 3 incubation days. Ribotyping profiles obtained for sub-samples collected after 5 and 7 incubation days were separated with confidence in different clusters, suggesting a response of microbial communities to H2 exposure. This discrimination was however transient as ribotyping profiles corresponding to the last incubation day could not be discriminated according to their respective H2 treatments (Fig. 4A). The transient response was likely due to the fact that H2 alone was not sufficient to maintain a sustained change in microbial community structure. Decrease of complex carbon sources and nutrients used along with H2 for mixotrophic growth in soil is a potential explanation for the convergence of the ribotyping profiles after 7 incubations days.

Figure 4 Influence of H2 exposure on bacterial ribotyping profile.

(A) UPGMA agglomerative clustering of soil samples derived from a matrix of Euclidean distance calculated after Hellinger transformation of OTU (97% identity) absolute abundance in soil microcosms exposed to eH2 and aH2 throughout a 10-day incubation period. The black circles ● represent significant nodes (P ≤ 0.05). (B) Principal component analysis showing the distribution of soil subsamples in a reduced space defined by the relative abundance of 16S rRNA gene sequences classified at the OTU level (97% identity). Only the OTUs (represented by red lines) having a higher contribution than average to explain the two dimensions of the PCA space are shown to facilitate visualization of the analysis. Soil subsamples collected in microcosms exposed to eH2 and aH2 are shown in blue and green, respectively.

A PCA was computed to identify OTUs contributing to the clusterization of ribotyping profiles. The ordination space defined by the first two components explained 38.6% of the variation observed (Fig. 4B). Three OTUs defined an important proportion of the reduced space represented by both axes (Fig. 5). The relative abundance of OTU 52 (classified as a member of the class Betaproteobacteria) contributed to distinguish the four soil subsamples collected before the incubation from the 20 other subsamples collected after 1, 3, 5, 7 and 10 incubation days along the first axis as it decreased after the beginning of the incubation period (Kruskal–Wallis, P < 0.05). Two OTUs classified as members of the order Bacillales (OTUs 65 and 299) contributed to discriminate both H2 treatments in the ordination space after 5 and 7 incubation days (Kruskal–Wallis, P < 0.05). These OTUs were more abundant in microcosms exposed to eH2 until day 10, where they reached a similar relative abundance in aH2 and eH2 treatments. The transient response of these OTUs in eH2 treatment could be a direct consequence of H2 exposure as representatives of the genus Bacillus possess putative membrane-bound type 1a and 1d [NiFe]-hydrogenases (Greening et al., 2016).

Figure 5 OTUs of interest to explain the PCA.

Time series of the relative abundance of the 3 OTUs having a higher contribution than average to explain the two dimensions of the PCA space in soil microcosms. The average and standard deviation measured in replicated microcosms are represented. The closest taxonomic affiliations of OTUs 52, 299 and 65 are, respectively, the bacterial order MND1 (Betaproteobacteria), genus Bacillus (Firmicutes) and the species Bacillus cereus (Firmicutes).

Conflicting results were obtained in two different studies reporting the impact of H2 soil exposure on bacterial community structure based on 16S rRNA terminal restriction fragment profiles analysis. H2 exerted no incidence on ribotyping profile in soil exposed to 250 nmol H2cm−3h−1 (500 ppmv H2 in artificial air, added at 45 ml min−1), while a single ribotype related to Mycobacterium increased in soils upon elevated H2 exposure (Osborne, Peoples & Janssen, 2010). On the other hand, an exposure rate of 33 nmol cm−3 h−1 (79 ppmv H2 at 100 cm3 min−1) exerted a significant influence on bacterial community profile, with an increase of T-RFLP peaks belonging to γ-Proteobacteria and a decrease of peaks belonging to Actinobacteria and α-Proteobacteria upon elevated H2 exposure (Zhang, He & Dong, 2009). A likelihood ratio analysis, fitting a negative binomial generalized log-linear model to the sequencing data (McCarthy, Chen & Smyth, 2012; Robinson, McCarthy & Smyth, 2010), unveiled that distribution of 958 OTUs was influenced by H2 exposure. The relative abundance of OTUs responding to H2 treatment ranged between 0.001 and 1.8%, suggesting an incidence of H2 on members of the rare biosphere and abundant taxa. The influence of H2 was uneven among different taxonomic groups as different representatives of the same taxa (i.e., OTUs classified at the phylum and the order taxonomic levels) were found to be favored or disfavored in response to eH2 exposure (Table 1). Even though the uneven response among taxonomic groups impairs prediction of H2 exposure on metabolic functions in soil (Langille et al., 2013), this observation is sufficient to demonstrate that previous investigations relying on low-resolution community fingerprinting techniques have considerably underestimated the response of soil microbial communities to H2 exposure.

Table 1 Summary of OTUs showing different relative abundance in eH2 and aH2 treatments (Likelihood ratio test, P < 0.05).

Altogether, 406 OTUs were more abundant in eH2 and 552 were more abundant in aH2. The eH2 and aH2 rows indicate the treatment in which the identified phylotypes are more abundant. A single or the two most abundant OTUs are identified for each phylum. A list of the 958 OTUs whose distribution was influenced by H2 treatments is provided in the raw data file accompanying the article.

Treatments	Phyla	Most abundant OTU (order level)	
eH2	Proteobacteria (27.6%)	Myxococcales	
Planctomycetes (10.6%)	Gemmatales, Phycisphaerales	
Bacteroidetes (9.6%)	Sphingobacteriales	
Chloroflexi (9.1%)	Ktedonobacteria	
Acidobacteria (7.1%)	Acidobacteriales, Solibacterales	
Verrucomicrobia (6.2%)	Verrucomicrobiales	
Actinobacteria (5.9%)	Actinomycetales, Solirubrobacterales	
Elusimicrobia (3.0%)	FAC88	
Gemmatimonadetes (2.7%)	Gemmatimonadales	
Firmicutes (1.5%)	Clostridiales, Bacillales	
Archaea (1.0%)	Methanobacteriales	
Armatimonadetes (0.7%)	CH21	
Chlorobi (0.2%)	SM1B09	
Others (14.8%)	–	
aH2	Proteobacteria (33.9%)	Rhodospirillales, Myxococcales	
Acidobacteria (11.4%)	Solibacterales	
Planctomycetes (11.1%)	Gemmatales	
Chloroflexi (6.2%)	Ktedonobacteria, A4b	
Bacteroidetes (5.1%)	Sphingobacteriales	
Actinobacteria (4.3%)	Actinomycetales, Acidimicrobiales	
Firmicutes (3.8%)	Clostridiales	
Verrucomicrobia (3.4%)	Spartobacteriales	
Gemmatimonadetes (2.7%)	Gemmatimonadales	
Archaea (1.8%)	SD-NA (Crenarchaeota)	
Elusimicrobia (1.4%)	Elusimicrobiales, MVP-88	
Armatimonadetes (1.3%)	Chthonomonadales, CH21	
Chlorobi (0.9%)	SM1B09	
Others (12.7%)	–	

Impact of H2 exposure on the co-occurrence of OTUs

Correlation networks were computed to investigate the impact of H2 exposure on the covariation of OTUs throughout the incubation period and identify OTUs for which the distribution is influenced by H2. Two separate networks were computed: the first for elevated H2 exposure (eH2 network) and a second for low H2 exposure (aH2 network). Both networks contained 43 modules and module preservation statistics demonstrated that module composition significantly changed across H2 treatments (Text S1). The significances of measured H2 oxidation rate in explaining network structure were 0.33 and 0.98 in aH2 and eH2 networks, again pointing out a significant impact of H2 on soil microbial community structure. Indeed, module eigengenes of five modules were significantly correlated with high-affinity H2 oxidation rate time series in eH2 network, while no module was related to the activity in aH2 network (Fig. S3). Together, these 5 modules represent 1,140 OTUs, which is more than a third of the whole eH2 network (3,154 OTUs). OTUs belonging to one module were hindered by eH2 exposure, while members of the four other modules were favored by eH2 treatment. The OTUs belonging to these five modules were members of the rare biosphere as well as more abundant ribotypes, with relative abundance ranging between 0.001% and 4.3% encompassing Proteobacteria, Chloroflexi, Acidobacteria as well as other phyla (Fig. S4). Clustering of these OTUs at the class and order levels unveiled that none of the taxonomic groups were restricted to the five modules correlated with H2 oxidation rate, supporting the previous observation that response to H2 exposure and the distribution of hydrogenase genes is uneven within each taxonomic group. The 38 modules in eH2 network whose eigengene showed no significant correlation with H2 oxidation rate are ecologically relevant observations since they represent indirect impacts of H2 exposure on soil microbial communities and highlight a previously overlooked role of H2 in shaping potential microbe-microbe cooperation and competition interactions. Deciphering the impact of these complex interactions on soil biogeochemical processes was beyond the scope of this study but will deserve attention in future investigations.

Conclusion

In conclusion, this exploratory study validated our hypothesis that elevated H2 exposure influences the activity of HOB, leading to direct impacts on the HOB but also indirect impact on the whole soil bacterial community through competitive and synergistic microbe-microbe interactions. Indeed, H2 soil exposure has an impact on the ecological niche of bacteria unevenly distributed among taxonomic groups beyond the alteration of HOB reported in early investigations. Our study was limited to a single farmland soil, with two replicated microcosms per treatment. Nonetheless, considering the significant impact that H2 exposure has brought to soil bacterial communities, it is reasonable to expect a similar effect on other soil types. Considering the steep H2 concentration gradient surrounding N2-fixing nodules, from 20,000 ppmv to sub-atmospheric levels along a 4.5 cm radius, the response of microorganisms might vary as a function of the distance from H2 diffusing sources as well as soil types, physicochemical conditions and the structural and functional structure of microbial community (Hartmann et al., 2015). The impact of H2 and HOB on soil microbial communities will definitely deserve more attention in soil microbiology. There is a need to further study whether elevated H2 exposure also exerts noticeable changes in soil ecological functions other than H2 oxidation, such as carbon metabolism, nutrient cycling, trace gas exchanges and xenobiotic degradation. Such a significant impact of a gas molecule on soil was observed in a study investigating the impact of a doubling atmospheric CO2 concentration, simulating climate change, on microbial communities (Zhou et al., 2011). A metagenomics approach on the matter might provide further answers on this rather ubiquitous microniche as well as the succession of hydrogenases along H2 concentration gradients in the environment.

Supplemental Information

Supplemental Information 1 Raw data

All raw data utilized for this manuscript are provided as supplementary material (H2 oxidation activities, soil physicochemical properties, list of OTUs whose distribution was affected by H2 treatments).

Click here for additional data file.

Text S1 Supplementary methods for correlation networks

Supplementary information for the computation of correlation networks.

Click here for additional data file.

Figure S1 Schematic representation of the dynamic microcosm chambers

(A) Schematic representation of the dynamic microcosm chambers utilized in this study to expose soil to aH2 or eH2 levels. (B) Photograph of one soil microcosm.

Click here for additional data file.

Figure S2 Taxonomic profiles in soil

Taxonomic profiles of the OTUs clustered at the phylum level for each microcosm at the beginning (day 0) and at the end of the incubation (day 10).

Click here for additional data file.

Figure S3 Module-Trait relationship heatmap for correlation network computed using OTU covariation profile under eH2 or aH2 exposure

Module-Trait relationship heatmap for correlation network computed using OTU covariation profile under eH2 or aH2 exposure. Each line in the heatmaps corresponds to a module. The colors in the heatmap stand for the Spearman correlation coefficient between the module eigengene and high affinity H2-oxidation rate measured in the microcosms. One module out of the 5 has the opposite sign (M1; red color in the heatmap eH2) because it is the only one showing a positive correlation (P < 0.05) with high affinity H2-oxidation rate. In contrast, the others modules (M2–M5; green color in the heatmap eH2) displayed a negative correlation (P < 0.05) with high affinity H2-oxidation rate.

Click here for additional data file.

Figure S4 Taxonomic profile of OTUs found in selected modules of correlation networks

Distribution of the relative abundance of the most abundant phyla in modules whose eigengene is significantly correlated with H2 oxidation rate in the eH2 network.

Click here for additional data file.

Table S1 Primers for sequencing bacterial 16S rRNA gene

List of the barcodes used to prepare bacterial 16S rRNA gene PCR amplicon libraries.

Click here for additional data file.

Additional Information and Declarations

Competing Interests

Author Contributions

DNA Deposition

Data Availability

The authors declare there are no competing interests.

Sarah Piché-Choquette conceived and designed the experiments, performed the experiments, analyzed the data, contributed reagents/materials/analysis tools, wrote the paper, prepared figures and/or tables, reviewed drafts of the paper.

Julien Tremblay and Susannah G. Tringe analyzed the data, contributed reagents/materials/analysis tools, reviewed drafts of the paper.

Philippe Constant conceived and designed the experiments, analyzed the data, contributed reagents/materials/analysis tools, wrote the paper, prepared figures and/or tables, reviewed drafts of the paper.

The following information was supplied regarding the deposition of DNA sequences:

Raw sequences were deposited in the Sequence Read Archive of the National Center for Biotechnology Information under the Bioproject PRJNA295403.

The following information was supplied regarding data availability:

Raw data an be found in the Supplemental Information.

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
