# Peer review of "H2-saturation of high affinity H2-oxidizing bacteria alters the ecological niche of soil microorganisms unevenly among taxonomic groups"

_PeerJ, doi:10.7717/peerj.1782_

## Round 0.1 · original submission · Major Revisions

· Academic Editor

Major Revisions

Dear dr. Piqué-Coquette,

Two expert reviewers have read your paper and on the basis of their pertinent comments you will need to make important revisions before the acceptance for publication.

Sincerely,

Maria Rosaria corbo

·

Basic reporting

1) Line 293: table S3 is reported in the text but there's no table S3 in the supplementary material. Please verify and correct
2) Figure S6 is reported in the supplementary material but not discussed in the text. Author should refer to it and discuss it if relevant to increase the information and discussion of the hypothesis.

Experimental design

No comments

Validity of the findings

no comments

Additional comments

1) in Line 121-122 a decrease in soil pH is reported, Contrary to author's statement, a drop from 5.9 to 5.1 could not be described as "slight". Even if this reduction was the same for the atmosferic- and eleveted-hydrogen microcosms, authors should at least discuss about this reduction in pH in batch microcosm epseriment and possible effect of pH on HOB.
2) in line 209-211 (and more briefly in the abstract), authors report that under eH2 an exponential significant decay of HAH population was observed. The figure S1 does not show any significant decay, since a reduction from about 3.5 to 2.5 E7 is far less than a exponential decay. Dealing with qPCR data, (based in gene copy whose number can differ by species and even strain, and can be biased by growth rate and DNA replication), it is adviced to be much more carefull in interpreting this result, even if statistically sound.
3) line 231, substitute "bacteria" with "H2 oxidation activity by bacteria"
4) LIne 235: please avoid the use of activity and activation in the same sentence

Reviewer 2 ·

Basic reporting

This manuscript by Piché-Choquette et al uses high quality genomic sequence data and network analysis tools to investigate the consequence of high H2 exposure to the H2 oxidation kinetics and soil microbial community. This is a fascinating topic that is highly relevant to questions of the roles of the microniche and local hotspots in soils to soil ecology and biogeochemistry. In particular, the study represents potential outcomes of concentrated H2 from N2 fixation nodules in the soils. This is a labor-intensive undertaking that has been analyzed and written up well.

In this study, the authors find consistent results with previous studies that exposure to high levels of H2 changes soil H2 uptake kinetics. The authors find very interesting trends in terms of changes to high affinity H2 uptake and complex changes in the microbial response that likely include specific members of the soil microbial community and indirect effects via microbe-microbe interactions. This study provides a valuable insight into the role of H2 in soil ecology, which will require future studies using metagenomic and other techniques to sort out in more detail. This study might benefit from moving some results from the supplement to the main text, especially to clearly illustrate the direct and indirect responses of the microbial community to H2 within the main text. Specific comments are below. This study is a valuable contribution to the field.

Comments:
Many of most significant figures in this study are hidden away in the supplement. The experimental setup is easily understood with the description in the text, and the main text of the paper does not need to include Figure 1 for the paper to stand alone without the supplement. I think it would be much more interesting to move Figure 1 to the supplement, and in its place, highlight the interesting results from one or more of the figures in the supplement (e.g., figures S 1,3,4,5).

I wonder if it would be worth defining the mole fractions associated with the different exposure levels (aH2, eH2) in Figure 1 and early in the text, and then relying on those aH2 and eH2 values, rather than always repeating the mole fraction in every legend and so often in the text? The reason to define aH2 and eH2 would seem to be to avoid inserting the concentrations over and over, and having many cases of redundancy.

L53: The terminology used to discuss hydrogenase classification should be referenced against recently published terminology by Greening et al., 2015a (already cited in this paper). In their paper, Greening et al classify the high affinity hydrogenase in a group 1 and compare with the terminology used in this paper: “The group 1h [NiFe]-hydrogenases have also been defined as the group 5 [NiFe]-hydrogenases; however, this work shows they are they are descended from other group 1 [NiFe]-hydrogenases.” I would suggest the authors make a similar acknowledgement of conflicting terminology for these hydrogenases in the literature and justify the reason for using the group 5 terminology if they decide to keep it. Especially relevant because other terminology is used later in this paper on L277.

L205: “while an increase of this uptake rate was measured in microcosms exposed to aH2 levels.” I think the wordage here makes it unclear if the authors are referring to the trend from hours 0 to 24, or from hours 24 to day 10. Please be more specific in the time period you mean in terms of days. The latter time period would be consistent with the data, but should be more clearly stated.

Figure 1: legend or text in figure itself could go into more detail on what CA, CB, HA, HB, eH2, and aH2 represent so the reader does not have to dig through the text. This is actually done in Figure 2, but I think it would be more appropriate to have in the legend of Figure 1.

Figure 2: Placement of H2 mole fractions in figure legend is confusing because this could be interpreted as the mole fraction that the H2 oxidation rate was quantified at. Sticking with eH2 and aH2 terminology would be less confusing.

Text in supplemental should be referred to as Text S1 in that supplemental file if referred to as such on L304

Figure S1 is as critical to the manuscript as Figure 2. Why not show both in main paper?

Figure S4: What do the colored blocks along the y-axis of each heatmap indicate? I find them distracting, and if they do not signify anything in particular, would number the or at least use grayscale so it doesn’t conflict visually with the heatmap. What is the significance of one module out of the 5 having the opposite sign?

Experimental design

L128-141: What tests were done to ensure that the method for measuring H2 fluxes was appropriate given the wide-ranging differences in H2 concentrations that the treatments were subjugated to? My concern is that soils incubated in 1000x higher H2 (H treatments) could degas H2 for a longer period than the 5-minute period that samples were flushed with synthetic air at 0.5 ppm. Residual degasing of H2 would cause measured H2 uptake rates to appear lower than they should and could be an alternate explanation for the trends observed in decreasing high affinity H2 oxidation rate for eH2 treatments (Figure 2). This could have been addressed with tests in sterilized or synthetic soils or by altering the flushing time. Were these or other tests done?

L142: I don’t find methods describing the qPCR results in figure S1.

L143: Here, or when the total mass of soil was presented on the previous page, the amount of soil removed for subsampling should be given for a sense of scale compared to the 200g sample. Was there any replacement, or just removal?

L210: More information on the statistical tests that arrive at the conclusion of a significant change in quantified hhyL quantity for eH2 but not aH2 should be given. This is not readily apparent from figure S1A. How was it determined to be an exponential decay?

L268 and Figure S3. The same statistical tests done to show the presence and absence of significant changes in hhyL (Figure S1) should be performed and reported for the data show in Figure S3. Are these significant differences over time and between treatments?

Validity of the findings

Agricultural soils can have marked changes to the soil microbial community. This could be addressed in the discussion with an appropriate citation.

L220: Does the lack of a relationship between hhyL DNA copies and elevated H2 necessarily indicate that HOB were not aided in persistence by H2? Depending on the duration and life phase that HOB are using H2, one might not necessarily expect that HOB can quickly change their hhyL copy number, or that they need to in order benefit from H2.

Additional comments

Short comments:

L30: ribotype instead of ribotypes

L32: The intended meaning of “H2
diffuse sources“ is not clear

L40: Would be useful to give the reader context on how 240,000L of H2 from N2 compares to H2 supplied by scavenging of atmospheric H2 on a volume/molar/mass H2 yr-1 basis.

L56: Is it worth using the HAH acronym – only used twice after defined.

L59: “evidence” not “evidences”

L79: “under saturating and saturating”. This language is not very precise.

L95: The citation to Khdhiri et al., 2015 should indicated similar methods were performed in that study, but not that it is a primary citation for the method.

L116: “microcosm” not “microcosms”

L 264: Long paragraph could be split

L 293: I do not see a table S3

---

## Round 0.2 · accepted · Accept

· Academic Editor

Accept

Dear Authors,

I have read your responses to the reviewers. It is a pleasure to accept your manuscript entitled "H2-saturation of high affinity H2-oxidizing bacteria alters the ecological niche of soil microorganisms unevenly among taxonomic groups" which you submitted to the PeerJ in its current form for publication in the PeerJ.